# Dream and Search to Control: Latent Space Planning for Continuous Control

## Abstract

Learning and planning with latent space dynamics has been shown to be useful for sample efficiency in model-based reinforcement learning (MBRL) for discrete and continuous control tasks. In particular, recent work, for discrete action spaces, demonstrated the effectiveness of latent-space planning via Monte-Carlo Tree Search (MCTS) for bootstrapping MBRL during learning and at test time. However, the potential gains from latent-space tree search have not yet been demonstrated for environments with continuous action spaces. In this work, we propose and explore an MBRL approach for continuous action spaces based on tree-based planning over learned latent dynamics. We show that it is possible to demonstrate the types of bootstrapping benefits as previously shown for discrete spaces. In particular, the approach achieves improved sample efficiency and performance on a majority of challenging continuous-control benchmarks compared to the state-of-the-art.

## 1 Introduction

Deep reinforcement learning (RL) has been effective at solving sequential decision-making problems with varying levels of difficulty. The solutions generally fall into one of two categories: model-free and model-based methods. Model-free methods (Haarnoja et al., 2018; Silver et al., 2014; Lillicrap et al., 2015; Fujimoto et al., 2018; Schulman et al., 2017) directly learn a policy or action-values. However, they are usually considered to be sample-inefficient. Model-based methods (Lee et al., 2019; Gregor et al., 2019; Zhang et al., 2018; Ha & Schmidhuber, 2018) learn the environment's dynamics. Common model-based approaches involve sampling trajectories from the learned dynamics to train using RL or applying a planning algorithm directly on the learned dynamics (Ha & Schmidhuber, 2018; Hafner et al., 2019a;b).

However, learning multi-step dynamics in the raw observation space is challenging. This is primarily because it involves the reconstruction of high-dimensional features (e.g., pixels) in order to roll-out trajectories, which is an error-prone process. Instead, recent work has focused on learning latent space models (Hafner et al., 2019a;b). This has been shown to improve robustness and sample efficiency by eliminating the need for high-dimensional reconstruction during inference.

**Learning on latent dynamics**: Once the dynamics have been learned, a classic approach is to sample trajectories from it to learn a policy using RL. This approach is usually motivated by sample efficiency. **Dreamer** (Hafner et al., 2019a) took this approach and demonstrated state-of-the-art performance on continuous control by performing gradient-based RL on learned latent dynamics. Another approach is to perform a look-ahead search - where the dynamics are used for a multi-step rollout to determine an optimal action. This could be accompanied by a value estimate and/or a policy that produces state-action mappings to narrow the search space or reduce the search's depth. **MuZero** (Schrittwieser et al., 2019) took this approach and applied tree-based search on latent dynamics - however, it was restricted to discrete action spaces only. The role of look-ahead search using learned latent dynamics has not been explored sufficiently for continuous action spaces.

**Our contribution**: In this paper, we extend the idea of performing look-ahead search using learned latent dynamics to continuous action spaces. Our high level approach is shown in Fig 1. We build on top of Dreamer and modify how actions are sampled during online planning. Instead of sampling actions from the current policy, we search over a set of actions sampled from a mix of distributions. For our search mechanism, we implement MCTS but also investigate a simple rollout algorithm

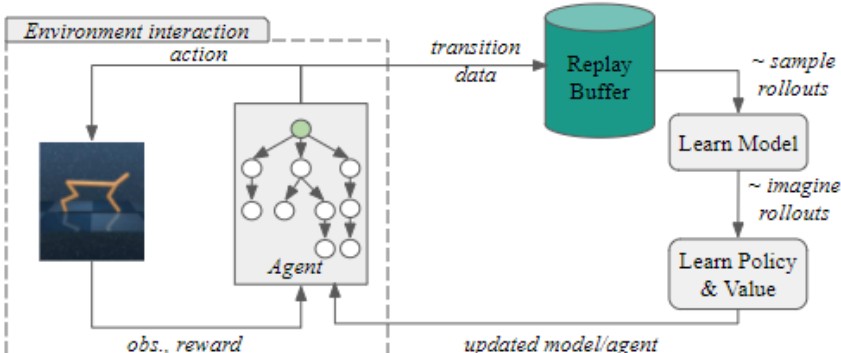

Figure 1: Overall Approach. Phase 1: At each time-step, determine the actions to be stepped via look-ahead search over latent dynamics and write the resulting transition data in a replay buffer. Phase 2: Sample fixed-horizon rollouts from replay buffer. Phase 3: Optimize latent model to reconstruct rewards and observations. Phase 4: Sample imaginary rollouts from the latent model to train agent.

that trades off performance for compute complexity. We observed that look-ahead search results in optimal actions early on. This, in turn, leads to faster convergence of model estimates and of optimal policies. However, these benefits come at the cost of computational time since deeper and iterative look-ahead search takes more time.

## 2 RELATED WORK

**Model-Based RL** involves learning a dynamical model and utilizing it to sample trajectories to train a policy or performing look-ahead search for policy improvement. World Model (Ha & Schmidhuber, 2018) learns a latent representation of observations and learns a dynamical model over this latent representation. It then utilizes the learned model to optimize a linear controller via evolution. The latent representations help to plan in a low dimensional abstract space. PlaNet (Hafner et al., 2019b) does end-to-end training of the latent observation encoding and dynamics and does online planning using CEM . Dreamer, the current state-of-the-art, (Hafner et al., 2019a) builds on top of PlaNet and uses analytic gradients to efficiently learn long-horizon behaviors for visual control purely by latent imagination. We built our work on top of Dreamer.

**Planning using look-ahead search:** Given a dynamics model, one can search over its state-action space to determine an optimal action. If one has a perfect model but a sub-optimal action-value function, then in fact, a deeper look-ahead search will usually yield better policies. This look-ahead search could be done both during online decision making and during policy improvement. AlphaGo (Silver et al., 2017; Lowrey et al., 2018) combined MCTS with a value estimator to plan and explore with known ground-truth dynamics. MuZero (Schrittwieser et al., 2019) learned a latent dynamic model and performed MCTS over it. They conditioned the latent representation on value and multi-step reward prediction rather than observation reconstruction. These works are limited to discrete action space due to large action space in continuous action space tasks.

A variation on MCTS is Progressive Widening (Coulom, 2006; Chaslot et al., 2008; Couëtoux et al., 2011) where the the child-actions of a node are expanded based on its visitation count. This has been exploited in continuous action spaces by adding sampled actions from a proposal distribution to create a discrete set of child-actions. AOC (Moerland et al., 2018) utilized this approach to apply MCTS in continuous action spaces over true dynamics. However, they showed their results only for the Pendulum-v0 task in OpenAI Gym. A common thread across all these prior works is the availability of known ground-truth dynamics. Another approach that can be used to reduce the size of the look-ahead search space is to utilize hierarchical optimistic optimization (HOO), which splits the action space and gradually increases the resolution of actions (Mansley et al., 2011; Bubeck et al., 2009).

## 3 DREAMER: LEARNING LATENT DYNAMICS AND POLICY

MBRL approaches usually involve collecting data from the environment, dynamics learning, and behavior learning by utilizing data collected from environment-agent interaction and/or trajectories generated via learned dynamics. We adopt Dreamer's dynamics and behavior learning approach. And, unlike Dreamer, we perform decision-time planning via look-ahead search during exploration to determine an action, instead of just using learned behavior policy. In this section, we review Dreamer's dynamics learning component in subsection (3.1) and the behavior learning component in subsection (3.2). After that, we describe our investigation of look-ahead search for decision-time planning in section (4).

### 3.1 DYNAMICS LEARNING

This comprises of a representation model that encodes the history of observations and actions to create a Markovian belief of the current state. This belief state and an action are consumed by a transition model that predicts the model's future latent state. The transition model is conditioned on an observation model and a reward model. The observation model predicts the observation corresponding to the current latent state and the reward model predicts the corresponding reward. The observation model is used only to condition the transition model and does not play a role during inference. These models are summarized n Eq. (1) where $(s_t, a_t, o_t, r_t)$ represent latent state, action, observation, and reward at discrete-time step $t$.

$$
\begin{aligned}
&\text{Representation model:} && p_\theta(s_t|s_{t-1}, a_{t-1}, o_t) \\
&\text{Observation model:} && q_\theta(o_t|s_t) \\
&\text{Reward model:} && q_\theta(r_t|s_t) \\
&\text{Transition model:} && q_\theta(s_t|s_{t-1}, a_{t-1}).
\end{aligned}
\tag{1}
$$

The transition model is implemented as a recurrent state-space model (RSSM) (Hafner et al., 2019b). The representation, observation model and reward model are dense networks. Observation features are given to the representation model, which are reconstructed by the observation model. This is unlike Dreamer which trains the model over raw pixels. The combined parameter vector $\theta$ in Eq. 1 is updated by stochastic backpropagation (Kingma & Welling, 2013; Rezende et al., 2014). It optimizes $\theta$ using a joint reconstruction loss ($\mathcal{J}_{\text{REC}}$) comprising of an observation reconstruction loss ($\mathcal{J}_{\text{O}}$), a reward reconstruction loss ($\mathcal{J}_{\text{R}}$) and a KL regularizer ($\mathcal{J}_{\text{D}}$), as described in Eq. (2).

$$
\begin{aligned}
&\mathcal{J}_{\text{REC}} \doteq \mathbb{E}_p\Big(\sum_t \big(\mathcal{J}_{\text{O}}^t + \mathcal{J}_{\text{R}}^t + \mathcal{J}_{\text{D}}^t\big)\Big) + \text{const} && \mathcal{J}_{\text{O}}^t \doteq \ln q(o_t|s_t) \\
&\mathcal{J}_{\text{R}}^t \doteq \ln q(r_t|s_t) && \mathcal{J}_{\text{D}}^t \doteq -\beta KL\Big(p(s_t|s_{t-1}, a_{t-1}, o_t) \quad || \quad (s_t|s_{t-1}, a_{t-1})\Big).
\end{aligned}
\tag{2}
$$

### 3.2 BEHAVIOR LEARNING

Given a transition dynamics, the action and value models are learnt from the imagined trajectories $\{s_\tau, a_\tau, r_\tau\}_{\tau=t}^{t+H}$ of finite-horizon ($H$) over latent state space. Dreamer utilizes these trajectories for learning behaviour via an actor-critic approach. The action model implements the policy and aims to predict actions that solve the imagination environment. The value model estimates the expected imagined rewards that the action model achieves from each state $s_\tau$, where $\tau$ is the discrete-time index during imagination.

$$
\begin{aligned}
&\text{Action model:} && a_\tau \sim q_\phi(a_\tau|s_\tau) \\
&\text{Value model:} && v_\psi(s_\tau) \approx E_{q(\cdot|s_\tau)}\sum_{\tau=t}^{t+H} \gamma^{\tau-t} r_\tau.
\end{aligned}
\tag{3}
$$

As in Eq. (3), action and value models are implemented as dense neural network with parameters $\phi$ and $\psi$, respectively. As shown in Eq. 4, the action model outputs a $tanh$-transformed Gaussian (Haarnoja et al., 2018). This allows for re-parameterized sampling (Kingma & Welling, 2013;

Rezende et al., 2014) that views sampled actions as deterministically dependent on the neural network output, allowing Dreamer to backpropagate analytic gradients through the sampling operation.

$$a_\tau = \tanh\big(\mu_\phi(s_\tau) + \sigma_\phi(s_\tau)\,\epsilon\big), \quad \epsilon \sim Normal(0, \mathbb{I}). \tag{4}$$

Dreamer uses $V_\lambda$ (Eq. 8) (Sutton & Barto, 2018), an exponentially weighted average of the estimates for different $k$-step returns for the imagination rollouts. The $k$ varies from $1 \ldots H$. It helps to balance bias and variance of target for value estimates by acting as an intermediary between the 1-step and the Monte-Carlo return. The objective for the action model $q_\phi(a_\tau|s_\tau)$ is to predict actions that result in state trajectories with high value estimates as shown in Eq. 5. It uses analytic gradients through the learned dynamics to maximize the value estimates. The objective for the value model $v_\psi(s_\tau)$, in turn, is to regress the value estimates as shown in Eq. (6). Also, Dreamer freezes the world model while learning behaviors.

$$\max_\phi \mathbb{E}_{q_\theta, q_\phi} \sum_{\tau=t}^{t+H} V_\lambda(s_\tau), \qquad (5) \qquad\qquad \min_\psi \mathbb{E}_{q_\theta, q_\phi} \sum_{\tau=t}^{t+H} \frac{1}{2}\big\| v_\psi(s_\tau) - V_\lambda(s_\tau))\big\|^2. \tag{6}$$

$$V_N^k(s_\tau) \doteq \mathbb{E}q_\theta, q_\phi \sum_{n=\tau}^{h-1} \gamma^{n-\tau} r_n + \gamma^{h-\tau} v_\psi(s_h) \quad \text{with} \quad h = \min(\tau + k, t + H), \tag{7}$$

$$V_\lambda(s_\tau) \doteq (1 - \lambda) \sum_{n=1}^{H-1} \lambda^{n-1} V_N^n(s_\tau) + \lambda^{H-1} V_N^H(s_\tau), \tag{8}$$

## 4 OUR APPROACH: DECISION-TIME PLANNING IN LATENT SPACES

Planning could be used to improve a policy in two ways (Sutton & Barto, 2018). **1) Background planning** involves optimizing a policy or value estimates with targets received from search or dynamic programming. **2) Decision-time planning** involves determining an optimal action to be stepped into the environment via search after encountering each new state. The simplest case would be to determine an action based on a current policy or value estimates. An alternate approach would be to perform a look-ahead search with the dynamics for determining an improved action.

In this work, we only focus on decision-time planning. We *hypothesize* that when we have imperfect action-value estimates during training, we could improve on that by look-ahead search in updated dynamics, leading to better action selection. These better actions lead to less noisy data in the replay buffer which leads to a better model and, in turn, a better policy. Under an oracle model, this is guaranteed to give a better policy than the current proposal policy. We implement two variations of decision-time planning: rollout and MCTS.

### 4.1 ROLLOUT

Rollout is based on Monte-Carlo Control and involves rolling out trajectories from a given state and action pair for a certain depth (or termination) in order to determine an action-value estimate. We sample actions based on the current policy as well as some other distribution (such as uniform/random) for better exploration from a given state. For each of the sampled actions, we perform rollout using current stochastic policy and estimate $V(\lambda)$ as in Eq. (8). A greedy action that maximizes the action-value estimates is selected and some noise is added to the greedy action during exploration. The computation time needed by rollout algorithm depends on the number of actions that have to be evaluated and the depth of the simulated trajectories. Upon action selection and environment interaction, we get a new observation based on which we re-estimate our belief state and re-plan for the next time-step.

Rollout is related to random shooting (Nagabandi et al., 2018) which generates multiple action sequences, computes a reward for each imagined trajectory and chooses the first action of the trajectory with the highest cumulative reward. It is also similar to Cross-Entropy method (CEM), which is an iterative procedure of sampling action sequences and fitting a new distribution to top-$k$ action sequences with the highest rewards. The main distinction of Rollout with these is that the actions are already sampled from a proposal distribution which helps to narrow our action search space.

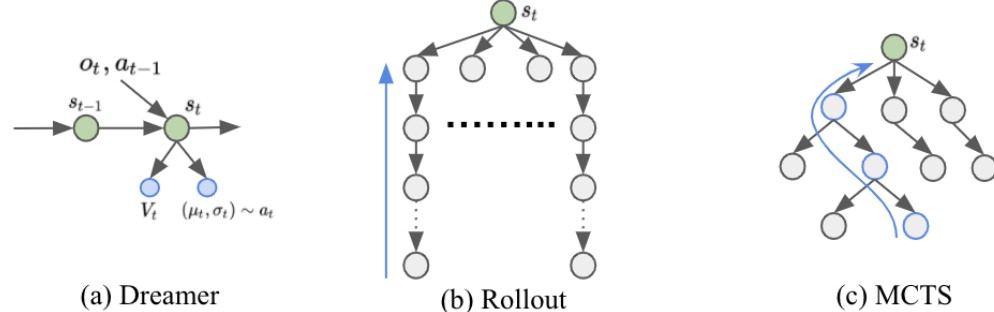

(a) Dreamer      (b) Rollout      (c) MCTS

Figure 2: Decision-time planning comparison: For a current state $s_t$, a) Dreamer simply samples an action using the current policy. b) Rollout samples $n$ actions and unrolls the dynamics for each action transition for a depth $d$ using mode action of current stochastic policy. c) MCTS iteratively extends a search tree via trade off over exploration and exploitation. In both (b) and (c), rollouts are used to update action-value estimates of the root edges (actions) and Value estimates of leaf nodes are used for bootstrapping. The blue line indicates path of backpropagted return for single rollout/simulation

## 4.2 MCTS

Unlike the Rollout algorithm which follows current policy for rollout, MCTS iteratively extends a search tree starting from a root node representing the current environment state. In this section, we describe how we adapt MCTS for continuous action spaces.

Every node of the search tree is associated with an internal state $s$. For each action $a$ from $s$ there is an edge $(s, a)$ that stores a set of statistics $N(s, a), Q(s, a), P(s, a), R(s, a), S(s, a)$, respectively representing visit counts $N$, mean value $Q$, policy $P$, reward $R$, and state transition $S$. Starting from the root node, MCTS executes the following three steps for a given computational budget:

**Selection**: Starting from a root node $s^0$, we traverse through the tree until we reach a leaf node $s^l$. At each time-step of traversal, an action $a^k$ is selected at each node. This is done by computing upper confidence bound (UCB) score (Rosin, 2011; Silver et al., 2018) for each child action(edge) using the stored internal statistics. The selected action is the one that maximizes this UCB.

***Continuous action.*** In continuous action spaces, we cannot enumerate all actions since there are infinitely many actions in a continuous space. An alternative could be to have a fixed number of actions for each node, where each action is sampled from a distribution. This fixed number could be pre-defined or determined via hyper-parameter search. This makes it similar to discrete-action MCTS.

Another alternative is to use *progressive widening*, where we slowly grow the number of child actions of state $s$ in the tree as a function of the total number of visits to that state $\{n(s) = \sum_{b \, \epsilon \, children(S)} N(s, b)\}$. Thus, if $| \, children(s)| < C_{pw}.n(s)^\alpha$ then, we add a new action to the parent node and expand that new action node for simulation. Otherwise , we simply select one of the actions from the current set of actions using an upper confidence bound. This was also done by (Moerland et al., 2018).

$$a^k = \arg\max_a \left[ Q(s, a) + P(s, a) \cdot \frac{\sqrt{\sum_b N(s, b)}}{1 + N(s, a)} \left( c_1 + \log \left( \frac{\sum_b N(s, b) + c_2 + 1}{c_2} \right) \right) \right] \quad (9)$$

The constants $c_1$ and $c_2$ are used to control the influence of the prior $P(s, a)$ relative to the value $Q(s, a)$ as nodes are visited more often.

**Expansion**: Upon reaching an un-expanded edge, we compute the leaf state $s^l = p_\theta(s^{l-1}, a^l)$ and it's reward $r^l = r_\theta(s^l)$ and a node corresponding to it is added to the tree. In the case of fixed actions set for each node, we sample $n$ actions and add them as edges to the node. Whereas, in the case of

progressive widening, no edges are added as they will be introduced upon re-selection during next simulation.

**Backup**: At the end of the simulation, the statistics along the trajectory are updated. The backup is generalized to the case where the environment can emit intermediate rewards, have a discount $\gamma$ different from 1, and the value estimates are unbounded. For $k = l...0$, we form an $l$-$k$ step estimate of the cumulative discounted reward, bootstrapping from the value function for $v^l$ as shown in Eq. (10).

$$G^k = \sum_{\tau=0}^{l-1-k} \gamma^\tau r_{k+1+\tau} + \gamma^{l-k} v^l \tag{10}$$

For $k = l...1$, we update the statistics for each edge $(s^{k-1}, a^k)$ in the simulation path as:

$$Q(s^{k-1}, a^k) := \frac{N(s^{k-1}, a^k) \cdot Q(s^{k-1}, a^k) + G^k}{N(s^{k-1}, a^k) + 1}; \qquad N(s^{k-1}, a^k) := N(s^{k-1}, a^k) + 1 \tag{11}$$

During action selection, we use normalized Q value estimates $\overline{Q} \in [0, 1]$ in the pUCT rule by using the eq. 12 as done by MuZero. This normalization is done by the minimum-maximum values observed so far in the tree.

$$\overline{Q}(s^{k-1}, a^k) = \frac{Q(s^{k-1}, a^k) - \min_{s,a \in Tree} Q(s, a)}{\max_{s,a \in Tree} Q(s, a) - \min_{s,a \in Tree} Q(s, a)} \tag{12}$$

## 5 EXPERIMENTS

In our experiments, we aim to determine the importance of search over latent space during exploration and, if that helps in policy improvement and better sample efficiency for control tasks, based on our hypothesis in section (4).

**Tasks.** We evaluated 20 control tasks from the DeepMind Control suite (Tassa et al., 2020), which are implemented in Mujoco. Each task's episode involves 1000 steps, and each step's reward is in the range $[0, 1]$. We use an action repeat of 2 across all tasks, as done by Dreamer. All episodes have random initial states. In all the considered tasks, we use the default feature representation of the system state, consisting of information such as joint positions and velocities, as well as additional sensor values and target positions where provided by the environment.

**Implementation.** We build our work on top of the existing Dreamer architecture and optimization with the change in way actions are selected during exploration. For the rollout case, we sample 100 actions from the proposal distribution and 50 actions from a uniform random distribution and perform rollout for 10 steps in latent space. These rollouts are performed as a batch operation giving us the freedom of choosing a large number of actions. The choice of large action count was motivated from (Van de Wiele et al., 2020). In the case of MCTS, we performed 50 simulations, and all child actions were sampled from the proposal policy distribution. This choice for the number of simulations was adopted from MuZero. We restricted ourselves from hyper-parameter search over range of simulations counts due to computational budget. In Dreamer, we added exploration noise from Normal(0, 0.3) to the mode action of the current stochastic policy during exploration. Furthermore, for rollout and mcts, the same exploration noise was added to all sampled actions as well as selected greedy action during exploration. Also, for MCTS, we used uniform prior for each edge and set $c_1 = 1.25$ , $c_2 = 19652$, $c_{puct} = 0.05$, $c_{pw} = 1$ and $\alpha = 0.5$

**Performance .** We treat Dreamer as our baseline, which already improves over model-free methods like DDPG and A3C in terms of sample efficiency as shared in (Hafner et al., 2019a). In the original paper, Dreamer operated directly on the pixel space. In this work, we adapt Dreamer to operate on the feature space provided by the environment. We summarize training curves for $2 \times 10^6$ steps in the figure (5) and show the mean performance across different environment interaction modes in figure (3a). Due to computational budget, we limited our experiments to only 2 million environment

steps to have a comparable comparison with MCTS experiments which require much longer training time as shown in figure (3b) due to synchronous nature of training.

In general, we observe that by incorporating look-ahead search during exploration, sample efficiency is improves. Furthermore, we also observe that the variance in learned policies' behavior is smaller than the baseline. It further improves as we replace the rollout search with the more complex MCTS look-ahead search. Also, In some cases, such as Hopper, we do observe that both search methods are slightly sample-inefficient compared to the baseline. One potential reason is that in the initial phase of training , the reward and value estimates' error may be high, which accumulates over look-ahead search. This could lead to bad updated estimates of action-values during decision-time planning, thereby it diverges from good actions. In both the cases of Hopper, search improves with training and eventually surpasses the baseline.

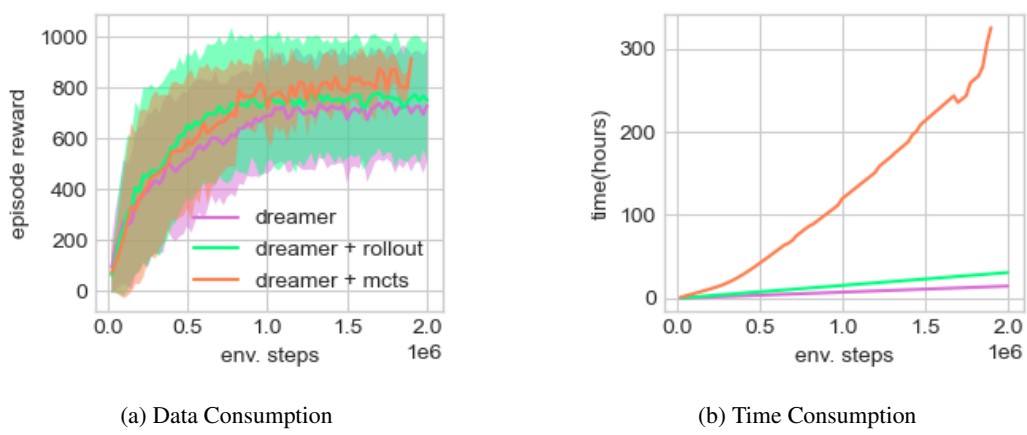

(a) Data Consumption                    (b) Time Consumption

Figure 3: a) Mean performance across 20 Benchmark tasks of dm-control suite. Each task was run for 3 random seeds. b) Time-taken by each explore method. MCTS is significantly slower than rollout due to the increased complexity of its search.

Notably, in most cases, we also observe that the performance of Rollout's matches that of MCTS. This implies that a complex and time-consuming look-ahead search may not be always required in the considered dm-control tasks , after a proposal policy has been learned. This is an interesting trade-off as MCTS takes computationally much longer than the rollouts which are performed as a batch operation.

While the above version of MCTS was implemented with progressive widening, we also investigated classic approach of simply having fixed children for each node.We considered 20 children for each node. We evaluated this implementation for 8 environments and show the mean performance across these tasks in fig. (4). The environments considered were Acrobat Swing Up, Cheetah Run, Reacher Hard, QuadrupedRun, Finger Spin, Hopper Hop, Walker Stand and Walk. Both the variants were given budget of 50 simulations.
across both the methods.

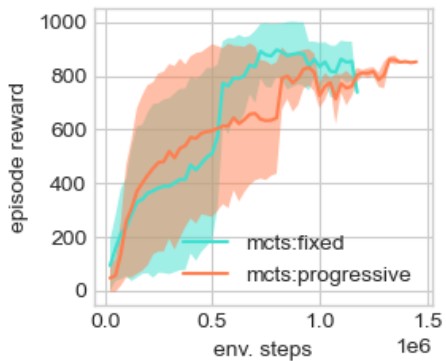

Figure 4: Comparison of mcts varitions - progressive widening and fixed children. We share mean scores across 8 environments trained on 3 random seeds.

We observe performance to be almost equivalent

## 6 SUMMARY & FUTURE WORK

In this work, we investigated the role of decision-time planning with the tree-based search over a learned latent dynamics for continuous control tasks during exploration. We show this leads to better sample efficiency in many tasks compared to the state-of-the-art Dreamer. The work can benefit from more complex tree-based search or better action selection expansion in tree. Also, we observe that the search component significantly increases the training time. As part of future work, we will explore the inclusion of distributional training for reducing computation time while still maintaining sample efficiency. Also, utilizing the output of the search as target estimates during optimization is another potential future thread. Finally, we will extend our investigation over more challenging tasks in terms of action-space, observation space, and complex goals.

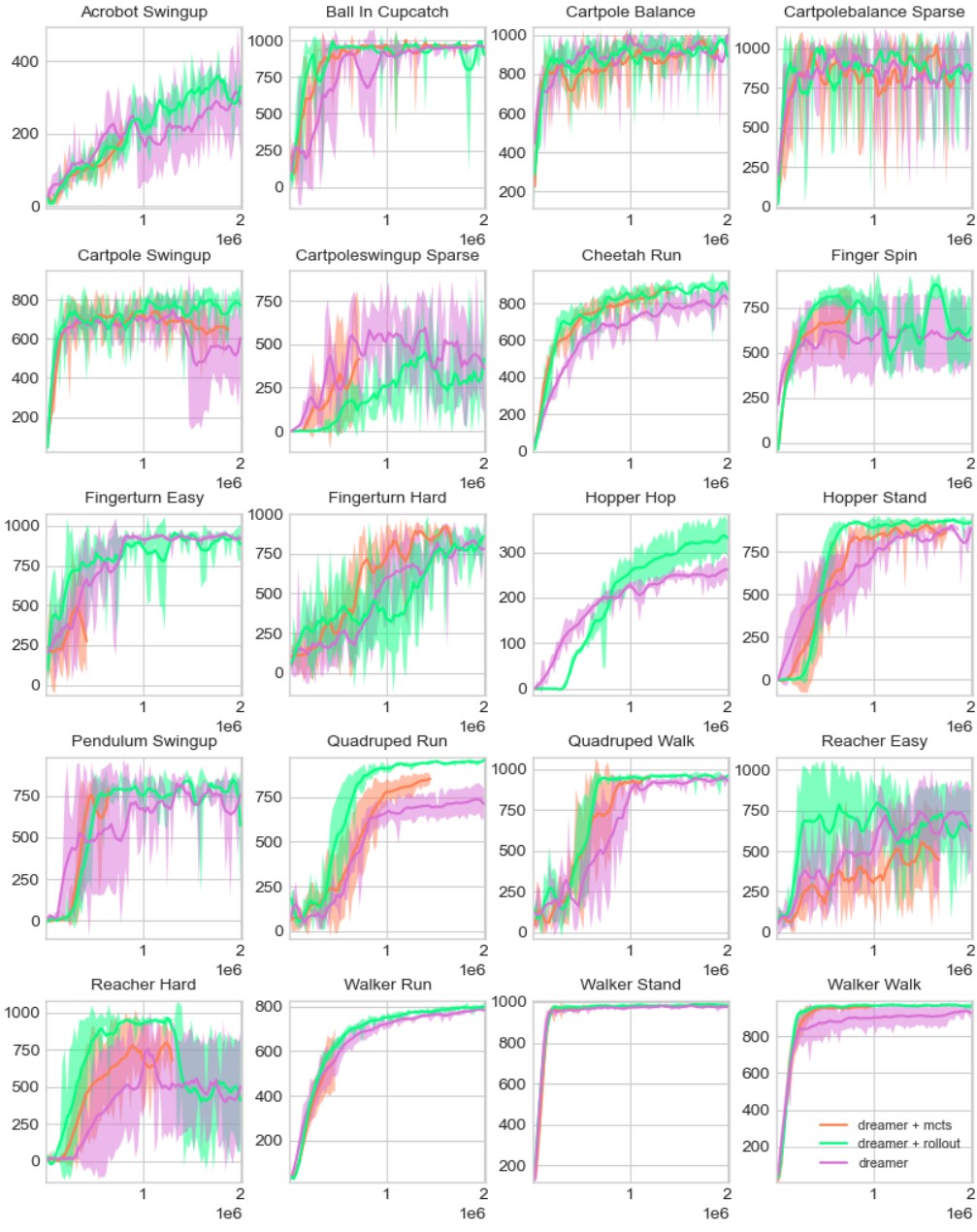

Figure 5: Comparison of Dreamer with different search modes during exploration. Each curve represents the mean value over 3 runs with different seeds.

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
