# OpenReview forum: "Dream and Search to Control: Latent Space Planning for Continuous Control"
_ICLR.cc/2021/Conference — Reject_

### Official Review · AnonReviewer4 · 2020-10-27

**Rating:** 5
**Confidence:** 4

**Review:**

The authors extend the Dreamer algorithm to use a different policy optimization mechanism, either a form of Monte-Carlo control or MCTS applied online in the continuous latent space at each decision step. This paper combines existing algorithms and compares two variants (“Rollout” vs “MCTS”) on continuous control domains.

Experimental results in 20 control tasks suggest that the proposed approaches and Dreamer perform similarly on most tasks. On some tasks, the “Rollout” and “MCTS” action-selection approach seems to have an advantage, but there is no clear trend showing that MCTS provides an advantage over the Monte-Carlo control in these cases. The number of simulations and rollouts were fixed for these experiments, so it prevents more nuanced interpretation of these results. Overall it’s not clear that the alternative action-selection mechanism for Dreamer proposed in the paper has any clear advantage overall. In the cases where there is an advantage (e.g. Hopper Hop, Quadruped Run), we don’t know whether the gains come from having a stronger policy improvement or from the modified behavior policy (which may help exploration), it would be interesting to investigate these questions in more detail.

Overall, the paper is clearly written and combines existing ideas in a sensible way, but it’s not clear what the take-away or potential impact of the paper is given there isn’t a particularly strong finding that comes out of this paper at the moment. Perhaps the authors could clarify their main takeaway for further discussion.

Additional questions:

* Is the policy prior for search updated based on the search policy (as in MuZero)?
* What network architecture was used for the experiments?

Minor things:


* Missing references for Progressive Widening technique in MCTS.
* It might be worth clarifying in the text that the tree policy is not actually a proper UCB, but is an approximation (PUCT which incorporates a prior policy).
* Return notation is inconsistent. G is used to denote random returns in Eq 10, but V is used in Eq 5.
* Missing q in last term of Eq2?
* The quality of the figures (resolution/format) should be improved
* is improves -> is improved

---

> ### Author Response · Authors · 2020-11-14
> **Thanks for your feedback | policy prior target explained | minor changes will be addressed**
>
> Thanks for your feedback.
>
> **`Is the policy prior for search updated based on the search policy (as in MuZero)?**
>
> Our method for updating the policy network differs significantly from MuZero, which uses the result of MCTS as supervision for the policy prior.
> In order to maximize sample efficiency, our optimization method follows Dreamer in generating a large number of imagined trajectories using the latent dynamics model. In order to use a Mu-Zero style update, we would have to run an MCTS search on each one of these trajectories. We found it computationally intractable to do so and therefore decided to use the faster analytic gradient method as done in Dreamer.
> The search method, whether rollout or MCTS-based, contributes to higher performance by increasing the quality of collected data that is used to train dynamics and serve the initial state for imaginary trajectories.
>
> **Network architecture:**
> The network architecture is identical to that used in Dreamer.
>
> **Minor points:**
>
> We will add a reference to the progressive widening technique.
>
> We already note in the paper that we are using a pUCT approximation. To make this fact clearer, we will also update section 4.2, in which we first introduce UCB action selection.
>
> We will make edits to fix the missing “q” term in Eq 2 and remove typos. The G term in Eq 10 refers to the n-step discounted return for a single pass, while the V in Eq 5 is the expected value.
>
> We will update the figures to make them clearer.

---

### Official Review · AnonReviewer2 · 2020-10-28
**weak novelty and experiments**

**Rating:** 4
**Confidence:** 4

**Review:**

summary:
This paper extends Dreamer, a model-based RL algorithm trained through latent imagination, by additionally performing decision-time planning in the learned latent-space dynamics for action selection. Most of the components follow those of Dreamer: from the experiences collected by the agent, it learns a (latent-space) world model that comprises representation model, observation model, reward model, and transition model, which are trained by minimizing reconstruction loss with a KL regularizer. The value and action models are also trained as same as Dreamer. It computes value estimates for the imagined trajectories and performs gradient ascent using the reparameterized gradient. Still, unlike Dreamer, the proposed method does not work on raw pixels but uses low-dimensional features such as joint positions and velocities. Finally, additional online planning, either a simple rollout or an MCTS, is performed to select an action at each interaction with the environment. Experimental results demonstrate that the proposed shows a better sample efficiency in several domains.



pros:
- This work shows that performing a search in a model-based RL can potentially benefit.

- Experimental results show that the proposed agent improves the sample-efficiency of the Dreamer baseline in several domains.


concerns:
- The main weakness of this work is the novelty. The difference to Dreamer is the additional adoption of online planning (or decision-time planning), but this additional component itself is not new to model-based RL.

- Planning in the learned latent-space dynamics is not well-motivated in the situation where the agent takes 'low-dimensional features' as an observation directly. Why does the agent have to learn the complex latent dynamics even when the low-dimensional features are accessible? Under this circumstance, it seems to be more natural to learn and plan on the dynamics in the raw (low-dimensional) observation space.

- It is not convincing that the proposed agent significantly outperforms the baseline. It is unclear what exactly the shaded area denotes in the plot (standard deviation? or standard error?), but the shaded areas of Dreamer and Dreamer+MCTS are being overlapped in Figure 3a.

- The analysis of why search is beneficial during exploration seems to be not substantial. More ablation experiments could have strengthened the paper. The raised hypothesis, that 'search can be beneficial when action-value estimation is imperfect', is not supported by evident ablation study. If the inaccurate action-value estimation is the problem, why (compounding) model error during a search should be less problematic?



comments and questions:
- In order to improve sample efficiency, structured exploration may be important. What aspect of the search of the proposed method can be helpful for better exploration (or other factors contributing to performance)?

- It seems that the dynamics of the continuous latent variables are stochastic. Since the number of latent states is infinite, there may be a need for special treatment to make MCTS tractable, but it is not clearly described. How was the stochastic transition handled in MCTS? (e.g. double progressive-widening is used?)

- Above Eq. (9): there is no definition of $\alpha$.

- The values of J in Eq. (2) should be maximized, not minimized, thus it seems to be awkward to be called a loss for each J. $q$ is omitted in the KL term.

- Experiments could have been more thorough. In Figure 5, not all the experiments were conducted until the 2 * 10^6 timesteps (e.g. dreamer+mcts in Quadruped Run, Reacher Hard, ...). It would be great to see ablation studies that show the effect of the planning horizon, search budget, and so on, which may be helpful to understand why the proposed method could perform better than the baseline. We can also explore which role of search is more significant between the search for exploration of collecting training samples and search at evaluation time.

- Other model-based RL algorithms that operate in the default feature representation could be a good baseline for comparison, e.g. MBPO (Janner et al. NeurIPS 2019).

- In figures 3-5, what does the shaded area represent? (standard deviation? standard error?)

---

> ### Author Response · Authors · 2020-11-14
> **Thanks for your review | novelty and importance of search elaborated**
>
> Thank you for your feedback. We provide our responses below for specific questions raised.
>
> **Novelty:**
>  We address this point in our overall comments above. The key contribution of our paper is to study the role of planning in latent space using a policy network as a proposal distribution for data-collection in control tasks.
>
> **Utility of planning in low-dimensional spaces is not well motivated. Why not learn in the raw dynamics space:**
> We do acknowledge that there might be specific environments where the raw state space is small enough that planning directly in that space is reasonable. However, since we experiment on a large suite of environments with varying state spaces, we chose to implement the latent representation as a common strategy. Also, planning in latent space can scale well across small and large state spaces. However, planning in the raw state-space usually does not scale beyond small state spaces.
>
> **Performance with respect to the baseline - overlapping shaded regions between Dreamer and Dreamer+MCTS:**
> The shaded regions represent standard error. The overlap is mainly due to the high standard error of the baseline Dreamer. We observe that the addition of planning actually provides a decrease in standard error. Further, we observe consistent improvement across environments: in 18 of 20 environments, our method outperforms Dreamer in terms of sample efficiency.
>
> **Analysis of why search is beneficial during exploration:**
> As you mention, we hypothesize that the gains shown by our method may arise because the learnt dynamics model is able to correct for errors in the policy network. This might enable the agent to gain access to optimal actions earlier in its data collection process. We agree that model learning errors might hinder this process, especially for long-horizon rollouts. However, model-based evaluation still provides enough refinement to improve overall performance. We will add this discussion to the paper.
>
> **What aspect of the search of the proposed method can be helpful for better exploration?**
> The core of our approach is to gain early access to optimal actions via search during data-collection. This leads to exploration of near-optimal trajectories, leading to better-learned dynamics and thereby leading to faster convergence to optimal policies. Investigating other forms of exploration is an interesting direction for future work.
>
>
> **Modification of MCTS to handle stochastic dynamics of the continuous latent variables:**
> As you mention, MCTS must be modified to deal with both continuous actions and latent space dynamics. We make action expansions at a node using two strategies: fixed and progressive widening. Once we have sampled an action, we sample a single next state from the latent dynamics. We have updated the paper to make a note of this.
>
> **Minor points:**
> $ \alpha \thinspace \epsilon \thinspace ]0,1[  $  is a hyperparameter of the MCTS search with progressive widening. We will add a description to the paper.
>
> We will update the paper to refer to “J” as objective.

---

### Official Review · AnonReviewer3 · 2020-10-29
**Adding MPC/MCTS to current Dreamer framework during training improves the final performance**

**Rating:** 6
**Confidence:** 4

**Review:**

##########################################################################

Summary:

The paper is developed on top of the Dreamer architecture, i.e. learning a latent space dynamics based on the image inputs to train policies. The difference is that, instead of using the already trained policy, this paper used MPC or MCTS to sample actions during the exploration phase to reduce the bias.  The authors demonstrated that their approach led to overall improved sample efficiency and final policy performance across many MuJoCo benchmark tests.

##########################################################################

Reasons for score:

Overall,  the methodology is sound and the writing is clear. The contribution, however, seemed minor since it is a small modification to the original Dreamer framework, and the improvement in the performance is not significant. Thus, my rating for this paper is weak acceptance.

##########################################################################

Pros:

(1) The writing is clear and easy to understand.

(2) Comprehensive studies on many experiments to study the effectiveness of their method, unlike many learning papers which only selected a small subset of validation tasks. This gives us a full picture of the strength and weakness of the proposed approach.

Cons:

(1) As mentioned before, the theoretic contribution of the paper is small. It modifies the SoTA with a minor tweak, and the results are not that significant.

(2) In this paper, the observations used are from the original state spaces of the environments. On the contrary, Dreamer assumes inputs in the image space and that is the reason a latent space was introduced. It remains to see if the claim still holds if the studies are executed in the pixel space.

##########################################################################

Conclusion:

Please add ablation studies in the image space as well and see if the conclusion still holds.

---

> ### Author Response · Authors · 2020-11-14
> **Thanks for your review| novelty elaborated | pixel space addressed**
>
>
> Thank you for your feedback. We provide our responses below for specific questions raised.
>
> **Novelty in comparison to Dreamer:**
> We address this point in our overall comments above. The key contribution of our paper is to study the role of planning in latent space using a policy network as a proposal distribution for data-collection in control tasks.
>
> **Pixel vs state-space observations:**
> We address this question in the general comments: we believe that the nature of the observation space is orthogonal to our contribution.

---

### Official Review · AnonReviewer1 · 2020-10-29
**Nice idea but unconvincing results**

**Rating:** 4
**Confidence:** 5

**Review:**

This paper proposes to integrate planning into Dreamer. The main idea is to apply a planning module on top of Dreamer to improve the quality of action selection. The planning via MCTS is on the learnt latent dynamics and the policy learnt by Dreamer. One of the challenges addressed in the paper is to perform planning on continuous action spaces. The proposed method is evaluated on 20 control tasks from the DeepMind Control suite, and compared against the original Dreamer algorithm, and a baseline planning method that does only rollout simulations.


The problem of using planning to enhance action selection for model-based RL is interesting and worth studying. However, the proposed idea is quite incremental. It might be worth trying, however, the experiment results are not exhaustive to evaluate the real benefits of the proposal.

In overall, the application of MCTS and the baseline Rollout on top of the deamer's learnt latent dynamics is quite straightforward. As the domain is continuous, therefore there is a bit challenge on the search tree's representation. Most techniques used in the paper is quite standard from existing works. In addition, there would be helpful if there are more ablation studies to look at the effect of the way MCTS is setup, .e.g. the amount of the fixed actions at branching, the number of simulations, etc. Those settings would affect how deep the policy tree is built, which roughly similar to the setting of horizon $H$ in Dreamer. The trade-off between this horizon length with the estimation error in model learning was well ablated in the Dreamer paper. It would be helpful to see the same ablation here. Given that MCTS can do planning under uncertainty (POMDP), i.e. on inaccurate model estimation, it would be great if the proposed idea discusses on this possibility and could address the problem of performance degradation with a large long look-ahead horizon.

The experiment results are not very convincing. There are many unfinished experiments. The proposed idea does not always outperform the baseline. A complex MCTS planning while consuming expensive computation, but performs worse than the baseline Rollout, and sometimes worse than the original Dreamer. More ablation studies might also be needed to make fair comparisons, i.e. while MCTS requires more planning time, could more computation budget be allocated for the baseline Rollout (more simulations or with larger tree settings) and Dreamer (more batch updates for action and value models)?


The notations, i.e. transitions, policy, etc., in Section 3 and 4 should be made consistent.

---

> ### Author Response · Authors · 2020-11-14
> **Thanks for your review | novelty, search comparison elaborated**
>
> Thank you for your feedback. We provide our responses below for specific questions raised.
>
> **The proposed idea is incremental:**
> We address this point in our overall comments above. The key contribution of our paper is to study the role of planning in latent space using a policy network as a proposal distribution for data-collection in control tasks.
>
> **More ablation studies on the MCTS set-up:**
> We also address this point in our overall comments above. Our primary goal was to demonstrate that our approach can improve sample efficiency without relying on significant tuning which is demonstrated by our results. The suggested ablations are indeed valuable - but would not affect the core takeaway of the paper.
>
> **Performance improvement with MCTS:**
> The reviewer comments that `“..a complex MCTS planning while consuming expensive computation, but performs worse than the baseline Rollout....” `
> First, we note that the MCTS-based agent actually outperforms the rollout-based agent, as can seen in Figure 3 and Figure 5. Second, the rollout-based agent is not proposed as a baseline but is instead a key contribution of our paper. The baseline for all our results is Dreamer only. While the MCTS based agent can be used to achieve maximum performance, the rollout-based method provides a simple and computationally efficient way in which planning can strengthen performance on continuous control tasks.
>
> **Incomplete runs:**
> As noted in the general comments above, a few experiments involving MCTS could not be completed in time due to a lack of computational resources. However, all results presented have been run for a sufficiently long period of time to establish that the relative performance of our methods. We continue to run the remaining few MCTS experiments and will update the final manuscript with extended graphs. We do not expect these results to change any of the conclusions in the paper.
>
> **Notations:**
> `“The notations, i.e. transitions, policy, etc., in Section 3 and 4 should be made consistent.”`
> It would be helpful if you could elaborate on these inconsistencies; we would be happy to make changes to clarify the manuscript.

---

### Author Response · Authors · 2020-11-14
**Overall Summary**

We thank the reviewers for their feedback and suggestions. In the following we share key points which address common concerns of some reviewers:

**```Key Novelty:** The key novelty is a method that uses planning in latent dynamics with a proposal distribution provided by a policy network to improve performance in continuous control tasks. We formulate two variants of this method and show that they both outperform the state-of-the-art baseline (Dreamer) in the majority of DMControl environments. Regarding the novelty of our approach, we are not aware of any work which investigates look-ahead search-based planning for continuous control with learned dynamics. In particular, Dreamer (Hafner et al 2019) and MBPO (Janner et al 2020) perform environment rollouts solely using the policy network. PlaNet (Hafner et al 2018) uses the model to choose actions solely using CEM.

**Incomplete MCTS runs:** Some reviewers noted that a few experiments involving MCTS were not conducted for the full 2M time steps. This was purely due to lack of computational resources. However, all experiments were run for a sufficient number of timesteps to establish the relative performance of our methods. We are continuing the remaining few MCTS runs and will update the final manuscript with extended graphs. We do not expect these results to change any of the conclusions in the paper.

**Pixel vs Non-pixel observations:**  Some reviewers pointed out that Dreamer uses image observations while we use state features. As we mention in our paper, for fair comparison, all results involving Dreamer utilize state features. This choice is dictated by computational considerations only. We reiterate that our objective is to perform planning with the help of learned dynamics irrespective of how those dynamics were learned. Our methodology would still work if the dynamics were trained in pixel space, which only requires minor changes in the architecture of the dynamics model.

**Ablations:** Another suggestion from reviewers was to ablate various aspects of the MCTS method such as the search depth or the number of expansions at a node. Our focus in this paper was to demonstrate the advantage of our method over prior work and show improved performance without a significant amount of tuning. Additional experiments may show further avenues for improvement; however, this is beyond the scope of this paper. We do, however, provide two important ablations of the key components of our method. First, we test both a rollout-based and an MCTS-based planning module and show that the rollout-based module captures much of the benefit of introducing planning. Second, we experiment with two different ways to modify MCTS to handle continuous action space, fixed sampling, and progressive widening, and show that the simpler fixed sampling option is adequate.

---

### Decision · Program_Chairs · 2021-01-07
**Final Decision**

**Decision:**

Reject

**Comment:**

This paper proposes an extension to the Dreamer agent in which planning (either via MCTS or rollouts) is used to select actions, rather than sampling from the policy prior. The results show small improvements over the baseline Dreamer agent.

Pros:
- Important study on incorporating decision-time planning into Dyna-based agents
- Evaluation on many control tasks rather than just a few

Cons:
- Lack of ablations and detailed analysis
- Claims aren't backed up by quantitative results

The reviewers generally felt that the approach taken in the paper lacked novelty. I agree that the approach is somewhat incremental (in fact I think it is also an instance of [1]). While both incremental changes and reimplementations of older methods with newer techniques can indeed be valuable, the current paper falls short in terms of the evaluation. As pointed out by several reviewers, there is no in-depth analysis explaining the design choices in which rollouts or MCTS are most likely to help (e.g. search budget, exploration parameters, etc.). As these parameters can play a large role in performance, I think it is important to characterize their effect on the agent---otherwise, I do not think there is a clear learning regarding how to translate these results to other domains and tasks. Additionally, and perhaps even more seriously, there are a number of claims made in the paper about the proposed method being more data efficient or higher performance. But, it is not clear visually that these improvements are statistically significant, and no quantitative tests have been run (and if the authors want to make a claim about data efficiency, I'd especially encourage them to report a metric like cumulative regret). Finally, while the incomplete runs are not a reason for rejection on their own, they do add to my overall sense that the paper is incomplete in its current form.

Given the above reasons, I do not feel this paper is ready for publication at ICLR. I'd encourage the authors to perform more careful ablations of the effect of incorporating search into the agent, and to back up their claims with more rigorous quantitative results.

One small point: the authors wrote in the rebuttal that "we are not aware of any work which investigates look-ahead search-based planning for continuous control with learned dynamics". Grill et al. [2] uses MCTS with learned dynamics in a modification of MuZero, though only applies it in one continuous control task (Cheetah Run).

1. Silver, D., Sutton, R. S., & Müller, M. (2008). Sample-based learning and search with permanent and transient memories. ICML.
2. Grill, J. B., Altché, F., Tang, Y., Hubert, T., Valko, M., Antonoglou, I., & Munos, R. (2020). Monte-Carlo tree search as regularized policy optimization. ICML.